# *EchoingPixels*: Aliasing-Resistant Joint Token Reduction for Audio-Visual LLMs

**Chao Gong** [1 2]  **Depeng Wang** [2]  **Zhipeng Wei** [3]  **Ya Guo** [2]  **Huijia Zhu** [2]  **Jingjing Chen** [1]

## Abstract

Audio-Visual Large Language Models (AV-LLMs) face prohibitive computational costs of processing massive, redundant audio-visual tokens. Existing unimodal compression techniques fail to capture the heterogeneous and mutually influential information density of joint audio-visual signals. Furthermore, we identify a fundamental and overlooked theoretical bottleneck in sparse token reduction: positional aliasing. We demonstrate that aggressive sparse sampling on standard position-encoded sequences violates the Nyquist limit relative to the effective token interval, causing phase-wrapping collisions that corrupt temporal monotonicity. To address this, we introduce *EchoingPixels*, a framework for aliasing-resistant joint token reduction. Our Cross-Modal Semantic Sieve performs extractive selection on the synergistic audio-visual stream, dynamically allocating budgets based on joint-modality saliency rather than fixed per-modality ratios. To resolve positional aliasing, we derive Sync-RoPE, a spectral low-pass filter for Rotary Positional Embeddings that adapts encoding bandwidth to the sparse sampling rate, preserving monotonic temporal relationships in the reduced stream. Experiments show that *EchoingPixels* achieves performance comparable to full models using only 5-20% of original tokens, validating theoretically grounded sparse learning as a robust solution for efficient AV-LLMs. Code is available at https://github.com/CharlesGong12/EchoingPixels.

## 1. Introduction

Audio-Visual Large Language Models (AV-LLMs) (Xu et al., 2025a;b; Wu et al., 2024; Zhang et al., 2023) have

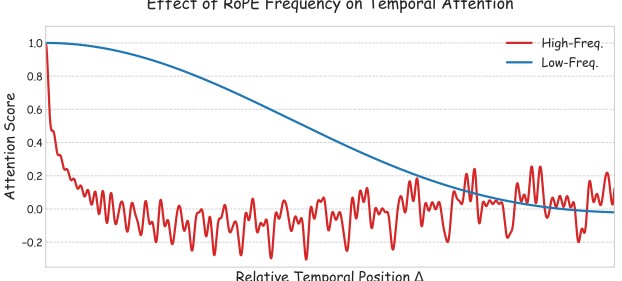

Figure 1. **Temporal RoPE dynamics.** High-frequency temporal encodings in vanilla AV-LLMs, cause attention to decay and oscillate over distance. In contrast, low-frequency components, as used in our Sync-RoPE, maintain stable signals.

demonstrated remarkable capabilities in understanding rich, multi-sensory inputs. However, their prowess comes at a steep price: the prohibitive computational overhead of processing massive token sequences from both video and audio streams. While token reduction has been explored for unimodal inputs, such as video-only models (Jiang et al., 2025; Yang et al., 2025a; Liu et al., 2025b; Li et al., 2024), its application to the audio-visual domain presents unique and unaddressed challenges. Simply compressing each stream independently is suboptimal, as it ignores the rich, synergistic relationships between sight and sound that are critical for genuine comprehension.

In this work, we identify a more fundamental and previously overlooked bottleneck that plagues any aggressive token reduction strategy: **Positional Aliasing**. We analyze and demonstrate that sparse sampling of token sequences, a cornerstone of many efficiency methods (Chen et al., 2024; Zhang et al., 2025a), can catastrophically corrupt the model's perception of time. Positional encodings like RoPE (Su et al., 2024) rely on a set of fixed rotational frequencies to represent sequence order. Powerful multimodal LLMs use high-frequency components in RoPE to model temporal information (Wang et al., 2024; Xu et al., 2025a; Bai et al., 2025b). When a sequence is made sparse, the effective temporal gap between adjacent tokens drastically increases. Therefore, large token gaps cause the high-frequency components to violate the Nyquist sampling limit (Shannon, 2006). This results in spectral folding, where distant timestamps become indistinguishable due to phase ambiguity,

---

[1]Fudan University [2]Ant Group [3]UC Berkley. Correspondence to: Huijia Zhu <huijia.zhj@antgroup.com>, Jingjing Chen <chenjingjing@fudan.edu.cn>.

*Proceedings of the 43rd International Conference on Machine Learning*, Seoul, South Korea. PMLR 306, 2026. Copyright 2026 by the author(s).

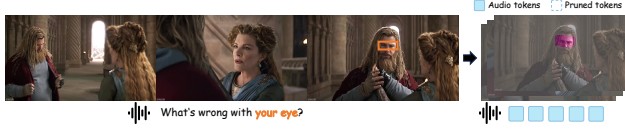

*(a)* Audio-guided saliency: The spoken phrase "your eye" guides the token budget to the character's eye.

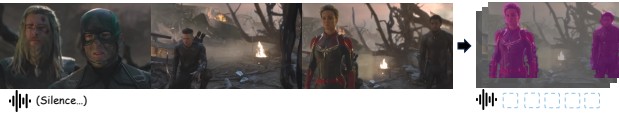

*(b)* Heterogeneous saliency: Ambient silence signals audio redundancy, guiding the token budget to the visual stream.

*Figure 2.* **The importance of cross-modal synergy for AV-LLM token reduction.** Selected tokens are highlighted.

fundamentally disrupting the model's temporal understanding (the red line in Figure 1). For AV-LLMs, this collapse of temporal integrity makes it difficult for the model to learn crucial relationships like audio-visual synchronization or event ordering.

To resolve this fundamental issue, we introduce *Echoing-Pixels*, a framework for aliasing-resistant joint token reduction. Our framework is built on two co-designed principles. First, to directly combat positional aliasing, we derive Synchronization-Augmented RoPE (**Sync-RoPE**). Acting as a spectral low-pass filter, Sync-RoPE mathematically adapts the positional encoding's frequency bandwidth to the sparse sampling rate by reallocating low-frequency channels to the temporal dimension. This ensures that the positional signal remains smooth and monotonic even across large temporal gaps, preserving the integrity of temporal relationships in the reduced sequence (the blue line in Figure 1).

Second, for Sync-RoPE to be effective, the model must first select a sparse yet semantically rich subset of tokens. This selection must be guided by cross-modal context, as the importance of any single token depends on the other modality in audio-visual scenarios (Figure 2a) or can be heterogeneous (Figure 2b). To this end, we propose the Cross-Modal Semantic Sieve (**CS2**), an extractive selection module. Unlike abstractive methods like Q-Formers (Li et al., 2023) that generate new summary tokens, CS2 operates on the unified audio-visual stream and selects the most salient original token representations from a single, combined pool. This extractive approach not only preserves the fine-grained details of the original signals but also allows for dynamic budget allocation between modalities based on their joint information density. Through extensive ablation studies, we validate our design choices for CS2, offering valuable insights into the optimal strategies for learning to select tokens in a multimodal context.

Our contributions are three-fold: (1) We are the first to

identify and systematically analyze the Positional Aliasing problem in sparse token sequences for AV-LLMs. (2) We design the *EchoingPixels* framework, which couples the aliasing-resistant positional encoding Sync-RoPE with an effective extractive selection module, CS2. (3) Extensive experiments show *EchoingPixels* achieves performance comparable to the full model using only 5-20% of tokens, validating that a theoretically-grounded approach offers a robust path toward efficient and powerful AV-LLMs.

## 2. Related Work

### 2.1. Audio-Visual LLMs

The landscape of Audio-Visual LLMs has been redefined by models such as GPT-4o (Hurst et al., 2024) and Gemini (Comanici et al., 2025), alongside open-source frameworks like Qwen-Omni (Xu et al., 2025a;b) and a growing body of work (Chen et al., 2023a;b; Wu et al., 2024; Lu et al., 2024; Fu et al., 2024; Cheng et al., 2024; Sun et al., 2024; Tang et al., 2025; Shu et al., 2025). The prevailing paradigm involves concatenating extensive sequences of visual and audio tokens, which, while effective, incurs prohibitive computational overhead. Despite this bottleneck, efforts to enhance efficiency remain incipient. Existing strategies are largely heuristic and unimodal, relying on static audio downsampling (Liu et al., 2025a), pruning based solely on instruction relevance (Zhong et al., 2025), or pure video pruning (Sun et al., 2025b). Crucially, these approaches treat audio and video modalities in isolation, failing to address the distinct and dynamic information densities of audio and video. Consequently, the challenge of performing fine-grained, interacted token reduction on the joint audio-visual stream—without sacrificing cross-modal synergies—remains an open problem.

### 2.2. Video Token Reduction

Video token reduction techniques generally fall into two paradigms: abstractive and extractive. Abstractive methods, epitomized by the Q-Former (Li et al., 2023), compress long sequences into a fixed set of learnable query tokens (Zhang et al., 2025b). While extended to audio-visual tasks (Sun et al., 2023; Yeo et al., 2025), its aggregation of each frame or window obscures precise positional information, risking the loss of fine-grained details (Yao et al., 2024; Huang et al., 2025b). In contrast, extractive methods maintain the fidelity of original features by selecting a subset of tokens. Training-free methods rely on calculating self-attention scores (Chen et al., 2024; Zhang et al., 2024a; Xing et al., 2024; Tao et al., 2025a; Yang et al., 2025b; Huang et al., 2025c) or similarities between frames (Tao et al., 2025a; Shao et al., 2025; Sun et al., 2025a; Hyun et al., 2025) to identify redundancy. To stabilize capabilities, learnable extractive strategies appear (Yang et al., 2025a; Jiang et al., 2025; Liu et al., 2025b).

Unlike prior video-only approaches that prune modalities in isolation, our framework operates on the joint audio-visual manifold. By preserving the positional information of the selected tokens, our extractive approach provides the necessary structural foundation for our spectral-aware Sync-RoPE mechanism to resolve positional aliasing.

## 2.3. Multimodal Positional Encoding

Rotary Position Embeddings (RoPE) (Su et al., 2024) have become the standard for injecting relative positional information in LLMs. In the multimodal domain, extensions like MRoPE (Wang et al., 2024) and TMRoPE (Xu et al., 2025a) partition the embedding dimension to independently encode time, height, and width. Recent theoretical studies (Huang et al., 2025a; Wei et al., 2025; Li et al., 2025a; Liao et al., 2025) analyze RoPE through a spectral lens, establishing that low-frequency components enhance long-video modeling. However, these optimizations operate under the common scenario of dense, continuous sequence. To our knowledge, no prior work has investigated the spectral implications of aggressive, non-uniform token sparsification.

## 3. Method

Our framework, *EchoingPixels*, is designed to solve the dual challenges of efficient token reduction and temporal integrity in AV-LLMs. Its core strategy is aliasing-resistant joint token reduction, realized by two co-designed components. First, to create a sparse sequence from the dense input, the Cross-Modal Semantic Sieve (CS2) performs extractive selection over the joint audio-visual stream. Second, to ensure the LLM can correctly interpret this sparse sequence, the Synchronization-Augmented RoPE (Sync-RoPE) remedies the positional aliasing artifacts introduced by aggressive token reduction. Figure 3 illustrates the overall architecture.

### 3.1. Cross-Modal Semantic Sieve

**Design Rationale: The Need for Holistic Context.** The primary goal of CS2 is to identify and select the most informative token representations from the combined audio and visual pool, along with the text input. Its design is motivated by three key requirements for effective cross-modal understanding and reduction.

*1. Cross-Modal Synergy.* The relevance of an event in one modality is often defined by its relationship to another. As shown in Figure 2, the visual of a character's eye becomes critically important when contextualized by the sound of "your eye". Any compression scheme operating on unimodal streams in isolation is blind to these synergies. Therefore, our sieve must operate on a joint audio-visual stream to capture these inter-dependencies.

*2. Instruction Pre-Fusion.* Reduction of audio and video

tokens leads to the loss of multimodal information. To preserve as much multimodal information in the LLM as possible, text tokens should anticipate and pre-fuse relevant information from all multimodal tokens before reduction. According to (Zhang et al., 2025b; Chen et al., 2024), this fusion stage implicitly occurs in the early layers of the LLM.

*3. Global Temporal Context.* Events in a sequence can gain significance retrospectively. An early, seemingly mundane visual frame might become crucial due to a later audio event. Standard causal LLM layers, where early tokens cannot attend to future ones, are incapable of such awareness. Furthermore, in typical AV-LLMs that concatenate modalities (Xu et al., 2025a; Tang et al., 2025; Liu et al., 2025a) (e.g., video-then-audio), early video tokens lack even concurrent audio context because they are concatenated before the audio. To make an informed reduction, each token must therefore be evaluated with access to the full, global context of the entire audio-visual sequence.

**Cross-Modal Encoder.** Based on this rationale, what we need for token representation is a cross-modal encoder $\mathcal{E}(\cdot)$ capable of cross-modal synergy, instruction pre-fusion, and bidirectional context. The early layers of LLM precisely meet the first two requirements; the third requirement can be met simply by replacing its causal self-attention with bidirectional self-attention. Therefore, the encoder $\mathcal{E}(\cdot)$ is a trainable copy of the first $N$ LLM layers. Let $T_v \in \mathbb{R}^{L_v \times D}$, $T_a \in \mathbb{R}^{L_a \times D}$, and $T_t \in \mathbb{R}^{L_t \times D}$ be the token sequences for video, audio, and text. We concatenate them into a single sequence $T = \text{Concat}(T_v, T_a, T_t)$, following the vanilla AV-LLMs approach. The sequence is then processed by our bidirectional encoder $\mathcal{E}(\cdot)$:

$$\hat{T} = \text{Concat}(\hat{T}_v, \hat{T}_a, \hat{T}_t) = \mathcal{E}(\text{Concat}(T_v, T_a, T_t)) \quad (1)$$

This process yields a set of deeply integrated representations $\hat{T}$, where each audio-visual token reflects its importance relative to the global context, and the text tokens have been fused with the multimodal information.

**Adaptive Extractive Selection.** Given the contextualized representations, we devise a strategy to select the most salient ones. We employ an extractive selection strategy, preserving a subset of the original representations rather than generating new ones like Q-Former (Li et al., 2023).

A learnable scorer, implemented as a two-layer MLP, projects each audio and visual representation $\hat{T}_i$ to a scalar importance score $s_i$, while text representations $\hat{T}_t$ are always preserved:

$$s_i = \text{MLP}(\hat{T}_i), \text{ for } i \in \{1, \dots, L_v + L_a\} \quad (2)$$

Based on a predefined total token budget ratio $r$, we apply a global Top-K operation across the unified set of $L_v + L_a$

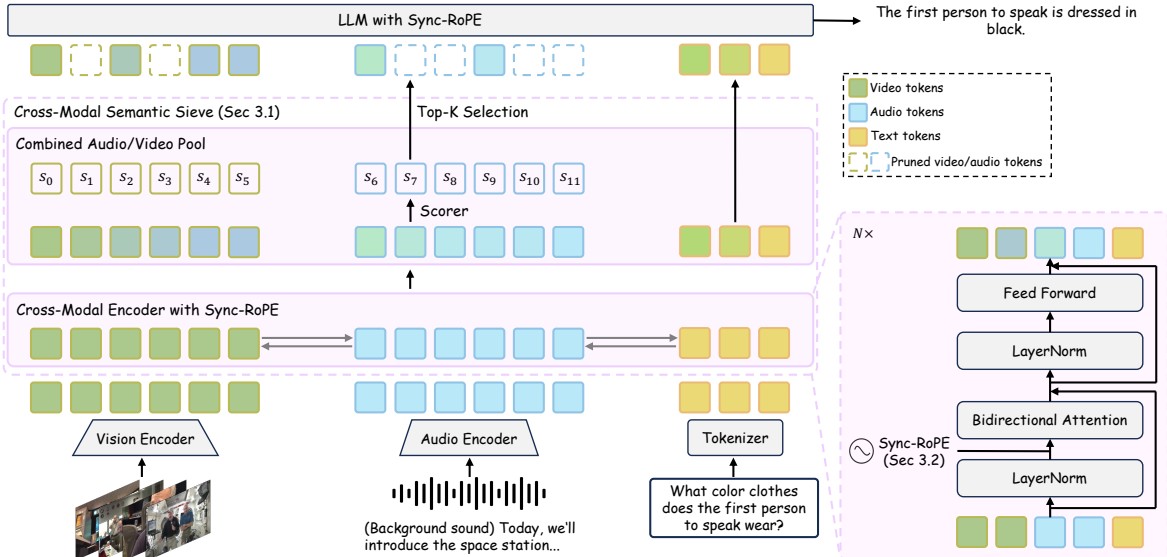

*Figure 3.* **The *EchoingPixels* Framework for Aliasing-Resistant Token Reduction.** Our framework processes the joint audio-visual stream in two stages. (1) The Cross-Modal Semantic Sieve (CS2) uses a bidirectional encoder to perform extractive selection, identifying the most salient token representations from a unified audio/video pool. (2) The resulting sparse sequence is then fed to the main LLM, where our Synchronization-Augmented RoPE (Sync-RoPE) positional encoding prevents aliasing artifacts and ensures robust temporal modeling. Sync-RoPE is also applied within the CS2 encoder.

audio-visual scores to identify the indices $I_{\text{sel}}$ of the top $k = r \cdot (L_v + L_a)$ representations:

$$I_{\text{sel}} = \text{TopK}(\{s_i\}_{i=1}^{L_v+L_a}, k), \tag{3}$$

$I_{\text{sel}}$ is then arranged according to the original token order, and the position indices of the selected tokens remain unchanged. The final compressed sequence fed to the main AV-LLM is $\text{Concat}(\{\hat{T}_i \mid i \in I_{\text{sel}}\}, \hat{T}_t)$.

This global, extractive approach provides two key benefits. First, operating on a unified audio-video pool enables dynamic budget allocation. The model is not constrained by fixed per-modality ratios and can flexibly allocate tokens to whichever modality is more informative for a given scene (e.g., more tokens for video during an action-heavy scene with silent ambient in Figure 2b). Second, as established, it preserves the original temporal and spatial positions of the selected tokens, a critical prerequisite for the aliasing-correction performed by Sync-RoPE in Section 3.2.

**Optimization with Surrogate Gradients.** The Top-K selection is a non-differentiable operation, which prevents end-to-end training of the MLP scorer via standard backpropagation. To address this, we employ the Straight-Through Estimator (STE) (Bengio et al., 2013), a surrogate gradient method. The mechanism operates as follows:

- Forward Pass: The hard 0/1 selection $y_i \in \{0, 1\}$ from Top-K for token $i$ is used and passed to subsequent layers.

- Backward Pass: While the true gradient $\partial y_i / \partial s_i$ is zero almost everywhere, the STE circumvents this by defining a surrogate gradient, effectively setting $\partial y_i / \partial s_i = 1$.

This creates a direct path for the gradient to flow back to the MLP scorer, effectively training it to assign higher scores to more useful tokens. As validated in our ablations in Section 4.4, STE provides a more stable and effective training signal for this sparse selection task compared to differentiable relaxations like Gumbel-Softmax (Jang et al., 2016).

### 3.2. Synchronization-Augmented RoPE

The CS2 module produces a sparse and irregularly timed sequence of representations. Feeding this directly to a standard AV-LLM would trigger the Positional Aliasing problem, as the model's high-frequency positional encodings are not designed for such large, non-uniform temporal gaps.

**Analysis of Positional Aliasing in Sparse Sequences.** CS2 yields a subsequence sampled from the original timeline with large and irregular index gaps. Let the retained token positions be an ordered set $P = \{p_1, p_2, \ldots, p_k\}$ where $p_1 < p_2 < \cdots < p_k$, and define the local effective stride as $\Delta_s(l) = p_{l+1} - p_l$. Standard AV-LLMs inject relative position $\Delta = n - m$ via RoPE (Su et al., 2024) by rotating queries/keys:

$$(R_{m,\Theta}\mathbf{q})^T(R_{n,\Theta}\mathbf{k}) = \mathbf{q}^T R(\Delta)\mathbf{k}, \tag{4}$$

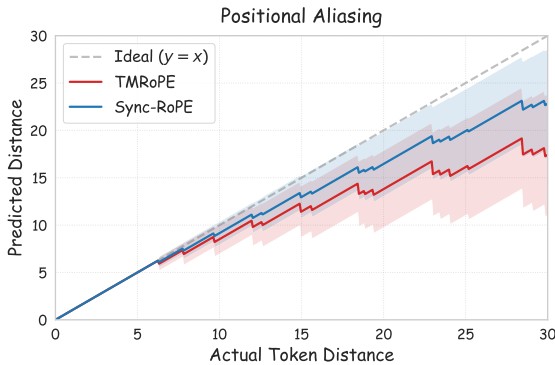

*Figure 4.* **Visualization of positional aliasing.** X-axis: Actual token distance; Y-axis: Predicted distance derived from RoPE. TMRoPE suffers from severe phase wrapping, mistaking distant tokens for adjacent ones (low predicted distance). In contrast, Sync-RoPE maintains near-ideal linear perception. See Section B.4 for the derivation of the predicted distance.

$$R(\Delta) = \bigoplus_{j=1}^{d/2} \begin{bmatrix} \cos(\Delta\theta_j) & -\sin(\Delta\theta_j) \\ \sin(\Delta\theta_j) & \cos(\Delta\theta_j) \end{bmatrix}, \quad (5)$$

where $m, n$ are the token positions, $d$ is the attention feature dimension, and $\theta_j = \theta_{\text{base}}^{-2(j-1)/d}$ denotes the rotation frequencies. TMRoPE (Xu et al., 2025a;b) for AV-LLMs further partitions the $d$ channels into $[t, h, w]$ to encode time, height, and width, assigning the highest frequencies (largest $\theta_j$) to the temporal partition for fine-grained dynamics.

Under sparsification, however, the model applies the same $\theta_j$ to time modeling while the effective sampling grid becomes significantly coarser. For a temporal frequency $\theta$, the phase increment between two adjacent retained tokens becomes $\Delta_s\theta$. When $\Delta_s\theta$ is large, the mapping $\Delta \mapsto R(\Delta)$ loses its distance-resolving property on the reduced grid. Specifically, there exist distinct relative distances $\Delta_1, \Delta_2$ with $|\Delta_1 - \Delta_2| \gg 0$ such that:

$$\Delta_1\theta \approx \Delta_2\theta \pmod{2\pi} \quad \Rightarrow \quad R(\Delta_1) \approx R(\Delta_2). \quad (6)$$

Consequently, distinct temporal separations yield indistinguishable attention logits $\mathbf{q}^T R(\Delta)\mathbf{k}$. We term this **Positional Aliasing**: aggressive downsampling forces high-frequency components into phase-wrapping collisions, creating ambiguity where distant tokens are mistaken for neighbors. This violation of temporal monotonicity corrupts the synchronization and causal ordering essential for AV tasks (Figure 4).

A conservative anti-aliasing condition can be derived from the Nyquist sampling theorem (Shannon, 2006). Summarizing the sparsity by a representative stride $T_s$ (e.g., the average of $\Delta_s$), preventing phase ambiguity requires that the phase shift between adjacent tokens remains within the

principal interval $[-\pi, \pi]$:

$$T_s\theta \leq \pi \quad \Leftrightarrow \quad \theta \leq \pi/T_s. \quad (7)$$

This condition is severely violated by the high-frequency temporal bands in standard TMRoPE under high compression ratios (see Section A for quantitative details).

**Sync-RoPE as a Spectral Low-Pass Filter.** To resolve the aliasing dilemma, we propose Sync-RoPE, reparameterizing the positional basis to satisfy the Nyquist criterion on the reduced grid.

Sync-RoPE repartitions the embedding dimension $d$ to explicitly separate the spectral bands into $[t_{\text{high}}, h, w, t_{\text{low}}]$, which is visualized in Figure 5. The *high-frequency* $t_{\text{high}}$ component is retained for compatibility with the pretrained model and for preserving local resolution for tokens that remain densely clustered (where $\Delta_s \approx 1$). Crucially, for the temporal backbone, we reallocate channels to an *ultra-low frequency* band $\Theta_{low}$. These frequency calibrations enable $\max(\Theta_{low}) \ll \pi/\mathbb{E}[T_s]$.

By enforcing this spectral bound, Sync-RoPE ensures that the phase shift between any adjacent retained tokens remains strictly within the principal interval $[-\pi, \pi]$. This mathematically guarantees a bijective and monotonic position-to-phase mapping globally, effectively applying a spectral low-pass filter that matches the positional encoding bandwidth to the sparse sampling rate and enhances model's time synchronization capability. We apply Sync-RoPE in both the CS2 encoder and the main LLM decoder, enhancing the entire system under aggressive token reduction.

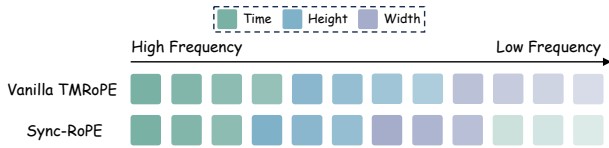

*Figure 5.* **Dimensionality partitioning for RoPE.** Unlike vanilla TMRoPE, which allocates only high-frequency channels for temporal information, Sync-RoPE repartitions the dimension to include both high- and low-frequency channels.

## 4. Experiments

### 4.1. Experimental Settings

**Models.** We evaluate *EchoingPixels* on two representative open-source AV-LLMs: Qwen2.5-Omni-3B and Qwen2.5-Omni-7B (Xu et al., 2025a). The Qwen2.5-Omni series utilizes a native dynamic resolution, which converts 28x28 pixel regions into tokens. For audio, it converts each second of audio into 25 tokens (each representing about 40ms).

**Training Data.** Our cross-modal encoder is initialized with shallow layers of the pre-trained LLM decoder. This re-

tains strong pre-trained capabilities and makes our training data-efficient. We use a total of only 390k samples. This comprises 123k LLaVA-Video samples re-annotated by Ola (Liu et al., 2025a) from both speech and visual perspectives, 37k AVQA samples (Yang et al., 2022) for scene sound, and 32k FortisAVQA samples (Ma et al., 2025) for music. Additionally, we include 200k samples with audio from the original LLaVA-Video dataset (Zhang et al., 2024b) to ensure sustained proficiency in video-centric tasks.

**Baselines.** We evaluate against five key comparators. (1) The Full Model, i.e., the uncompressed Qwen2.5-Omni. (2) IntraModal, a baseline established by combining strong unimodal compression schemes. For video, we use pixel un-shuffle adopted by leading VLLMs (Bai et al., 2025b; Wang et al., 2025), which merges each $2\times2$ token block into one token followed by an MLP projection. For audio, we use temporal convolution to merge 4 adjacent tokens, a common and effective strategy in powerful audio models (Das et al., 2024; Li et al., 2025b). (3) FastV (Chen et al., 2024), a representative attention-score-guided compression method. It is incompatible with efficient attention implementations (Dao, 2023; Wolf et al., 2020). (4) PyramidDrop (Xing et al., 2024), dropping tokens in a pyramid-like style based on attention score. (5) OmniZip (Tao et al., 2025b), a concurrent work on audio-video tasks that uses audio to guide video reduction thus cannot handle video-only scenarios.

**Benchmarks.** We evaluate on Daily-Omni (Zhou et al., 2025b), WorldSense (Hong et al., 2025), and Video-MME (Fu et al., 2025) (with audio) for audio-visual understanding, and Video-MME (without audio) and MLVU-dev (Zhou et al., 2025a) for video-only evaluation. All methods are evaluated under identical settings; WorldSense/Video-MME/MLVU use VLMEvalKit (Duan et al., 2024), and Daily-Omni follows its official code.

**Evaluation Metrics.** In addition to the metrics of all benchmarks, we also report an efficiency analysis, presented in terms of inference latency and CUDA memory.

**Implementation Details.** We report token budgets of 5%, 10%, 20% on Qwen2.5-Omni-3B and 10% on Qwen2.5-Omni-7B. CS2 uses $N=4$ layers, and Sync-RoPE partitions the 64 available frequencies into a [18,18,18,10] split. We fine-tune CS2 and the decoder for 1 epoch with batch size 128 and learning rate $2 \times 10^{-5}$. IntraModal is trained with same data.

### 4.2. Main Results

We present the main results of *EchoingPixels* in Table 1. Notably, Qwen2.5-Omni demonstrates leading performance on key benchmarks (e.g., 46.1 on WorldSense, 60.48 on Daily-Omni), outperforming other models like Vita-1.5 (36.9 on WorldSense) and Ola (50.71 on Daily-Omni), thus repre-

senting a strong and advanced open-source AV-LLM. Our analysis reveals that *EchoingPixels* not only achieves significant efficiency gains but also consistently outperforms other compression strategies across a range of token budgets.

A primary observation is the remarkable ability of *Echoing-Pixels* to maintain high performance with only a fraction of the original tokens. On the Qwen2.5-Omni-3B model, our method retains an impressive 99.0% of the full model's relative performance while using just a 20% token budget. This trend holds for the larger 7B model, where *Echoing-Pixels* at a 10% budget preserves 94.1% of the baseline's performance, validating its scalability.

The superiority of our cross-modal approach becomes evident when compared to the identically trained IntraModal baseline. This baseline, which compresses audio and video streams independently, suffers a substantial performance degradation, dropping to 84.2% on the 3B model despite a lenient 25% budget. In stark contrast, *EchoingPixels* with same training data and a stricter 20% budget vastly outperforms it. This performance gap underscores a core thesis of our work: leveraging cross-modal synergies through joint-stream processing is critical for effective audio-visual token reduction.

Our framework also demonstrates clear advantages over other token reduction paradigms. FastV and PyramidDrop, two representative attention-score-guided methods, not only lag in performance but also prove impractical for long-context tasks. Their incompatibility with efficient attention mechanisms leads to Out-of-Memory (OOM) errors on the 7B model. Similarly, while the concurrent work OmniZip shows competitive performance on audio-visual tasks, its reliance on audio to guide video token selection renders it incapable of handling video-only inputs, as indicated by its N/A results. *EchoingPixels* offers both computational stability and versatile modality support.

### 4.3. Efficiency Analysis

We analyze the efficiency of *EchoingPixels* by measuring inference latency and peak CUDA memory usage on Daily-Omni across different token budgets, as presented in Table 2. Our analysis focuses on the overall LLM forward latency (including our CS2 module), with a detailed module latency analysis provided in Section C.3. The videos for the Daily-Omni benchmark are 30-60 seconds long, and the batch size is 1.

In terms of latency, *EchoingPixels* delivers significant acceleration. With a 20% token budget, we achieve a 2.23x speedup in forward latency. The speedup also increases as the token budget decreases, reaching 2.96x at 5%. This is a result of applying a smaller sequence to an $O(L^2)$ complexity attention mechanism. These results take into account the

*Table 1.* **Main results on audio-visual and video-only benchmarks.** "Rel. Perf." shows the average performance relative to the Full Model. "OOM" indicates Out-of-Memory errors and "N/A" indicates incompatibility with the task. *EchoingPixels* consistently outperforms the baselines and retains $> 94\%$ of the Full Model's performance at a 10% budget.

| Model | Token Budget | Audio-Visual Tasks | | | | Video Tasks | | | Rel. Perf. |
|---|---|---|---|---|---|---|---|---|---|
| | | WorldSense | Daily-Omni | Video-MME w/ audio | Avg | Video-MME w/o audio | MLVU | Avg | |
| *Qwen2.5-Omni-3B* | | | | | | | | | |
| Full Model | 100% | 45.4 | 59.65 | 63.1 | 56.1 | 60.9 | 67.2 | 64.1 | 100.0% |
| IntraModal | 25% | 37.4 | 46.20 | 52.1 | 45.2 | 51.4 | 63.4 | 57.4 | 84.2% |
| FastV | 20% | 40.1 | 49.87 | 51.9 | 47.3 | 51.8 | 56.3 | 54.1 | 84.6% |
| PyramidDrop | 20% | 39.1 | 52.97 | 56.2 | 49.4 | 53.9 | 59.7 | 56.8 | 88.3% |
| OmniZip | 20% | 39.5 | 50.63 | 57.9 | 49.3 | N/A | N/A | N/A | 87.9% |
| *EchoingPixels* | 20% | 45.0 | 60.65 | 60.7 | 55.5 | 58.6 | 68.3 | 63.5 | **99.0%** |
| *EchoingPixels* | 10% | 43.5 | 57.56 | 58.4 | 53.2 | 55.4 | 67.4 | 61.4 | 95.2% |
| *EchoingPixels* | 5% | 40.9 | 52.88 | 55.7 | 49.8 | 54.7 | 66.1 | 60.4 | 91.0% |
| *Qwen2.5-Omni-7B* | | | | | | | | | |
| Full Model | 100% | 46.1 | 60.48 | 66.3 | 57.6 | 63.6 | 68.4 | 66.0 | 100.0% |
| IntraModal | 25% | 40.6 | 49.79 | 56.5 | 49.0 | 55.1 | 65.0 | 60.1 | 87.6% |
| FastV | 20% | OOM | 53.55 | OOM | 53.6 | 54.6 | 59.4 | 57.0 | 87.1% |
| PyramidDrop | 20% | OOM | 52.97 | OOM | 53.0 | 57.1 | 60.1 | 58.6 | 88.0% |
| OmniZip | 20% | 41.1 | 55.39 | 62.8 | 53.1 | N/A | N/A | N/A | 91.8% |
| *EchoingPixels* | 10% | 47.4 | 56.98 | 64.1 | 56.2 | 53.8 | 62.9 | 58.4 | **94.1%** |

*Table 2.* **Efficiency analysis on Qwen-Omni-3B.** We report the speedup of the average forward pass and the reduction in peak CUDA memory, calculated relative to the 100% baseline.

| Model | Token Budget | Forward Latency (ms) | Speedup (x) | CUDA Mem (GB) | Mem. Reduction (x) |
|---|---|---|---|---|---|
| Qwen2.5-Omni-3B | 100% | 517.1 | 1.00 | 32.03 | 1.00 |
| *EchoingPixels* | 20% | 231.7 | 2.23 | 14.15 | 2.26 |
| *EchoingPixels* | 10% | 194.0 | 2.67 | 13.10 | 2.45 |
| *EchoingPixels* | 5% | 174.8 | 2.96 | 12.26 | 2.61 |

*Table 3.* **Ablation on the modality synergy in the CS2 encoder.** We analyze the impact of bidirectional attention (vs. Causal and Intra-Modal) and text pre-fusion. "Full Synergy" refers to our proposed design.

| Methods | AV Event Alignment | Comparative | Context Underst. | Event Sequence | Inference | Reasoning | Overall |
|---|---|---|---|---|---|---|---|
| Causal Attention | 47.90 | 58.78 | 51.30 | 51.63 | 70.13 | 65.71 | 56.06 |
| Intra-Modal Attention | 49.58 | 67.18 | 52.33 | 51.63 | 68.83 | 68.57 | 57.73 |
| w/o Pre-fusion | 47.48 | 65.65 | 53.37 | 52.94 | 67.53 | 68.00 | 57.39 |
| Full Synergy | 52.10 | 61.83 | 59.07 | 57.19 | 72.73 | 68.57 | **60.65** |

overhead of our CS2 module, proving our framework makes an excellent efficiency trade-off. The advantages are also significant in terms of memory. *EchoingPixels* achieves a 2.26x reduction in peak CUDA memory at 20%, scaling to 2.61x at a 5% budget. Such substantial savings are crucial for resource-constrained scenarios and larger batch sizes during training.

### 4.4. Ablation Study

To validate the design choices of EchoingPixels, we conduct a series of comprehensive ablation studies on the Daily-Omni benchmark, with all experiments based on the Qwen-Omni-3B model at a 20% token budget with the same training data. The detailed category definitions of Daily-Omni are attached in Section B.3. We systematically dissect the contributions of our core components: the modality synergy within the CS2 encoder, the token selection strategy, the frequency partitioning of Sync-RoPE, and the ablation on CS2 encoder depth.

**Analysis of Cross-Modal Encoder Design.** We first investigate the critical design elements of the CS2 encoder that enable holistic context modeling, with results shown in Table 3. Our full model, which leverages bidirectional cross-modal attention and text pre-fusion, achieves the best overall performance. Replacing bidirectional attention with a standard Causal Attention mechanism leads to a substantial 4.59-point drop, confirming our hypothesis that retrospective, global context is vital for identifying salient events. Similarly, restricting the model to Intra-Modal Attention, where each modality only attends to itself, results in a 2.92-point performance decrease. This highlights the inadequacy of isolated processing and validates the necessity of joint-stream interaction to capture cross-modal synergies. Finally, removing the text instruction pre-fusion (w/o Pre-fusion) also degrades performance, demonstrating that providing multimodal information early in the process is crucial for understanding. These results collectively affirm that the full synergy of our CS2 design is essential for its success.

*Table 4.* **Ablation on token selection strategies.** Our learnable scorer with STE is compared against heuristic (Random, Similarity), optimization (Gumbel-Softmax), and budget allocation (Per-modality) alternatives.

| Methods | AV Event Alignment | Comparative | Context Underst. | Event Sequence | Inference | Reasoning | Overall |
|---|---|---|---|---|---|---|---|
| Random | 50.00 | 58.78 | 53.37 | 56.54 | 67.53 | 66.29 | 57.81 |
| Similarity | 32.35 | 50.38 | 38.34 | 41.50 | 48.05 | 47.43 | 41.85 |
| Gumbel-Softmax | 50.00 | 61.83 | 54.40 | 56.86 | 71.43 | 67.43 | 59.06 |
| Per-modality | 52.52 | 62.60 | 53.89 | 54.90 | 66.23 | 66.29 | 58.23 |
| STE | 52.10 | 61.83 | 59.07 | 57.19 | 72.73 | 68.57 | **60.65** |

*Table 5.* **Ablation on Sync-RoPE frequency partitioning.** The format is $[t_{\text{high}}, h, w, t_{\text{low}}]$. Our final selection is [18,18,18,10].

| Model | Frequency Split | AV Event Alignment | Comparative | Context Underst. | Event Sequence | Inference | Reasoning | Overall |
|---|---|---|---|---|---|---|---|---|
| Full Model | [16,24,24,0] | 50.84 | 66.41 | 54.40 | 53.27 | 77.27 | 68.00 | 59.65 |
| Ours | [16,24,24,0] | 48.58 | 60.30 | 53.89 | 54.58 | 69.48 | 68.00 | 57.98 |
| Ours | [20,20,20,4] | 50.84 | 62.60 | 55.44 | 55.88 | 71.43 | 70.29 | 59.65 |
| Ours | [16,16,16,16] | 48.74 | 62.60 | 54.40 | 57.84 | 72.08 | 68.00 | 59.31 |
| Ours | [18,18,18,10] | 52.10 | 61.83 | 59.07 | 57.19 | 72.73 | 68.57 | **60.65** |

**Analysis of Token Selection Strategies.** We examine the effectiveness of our extractive selection mechanism against several alternatives, as detailed in Table 4. Our chosen approach, using a learnable scorer with a Top-K operation trained via STE, sets the highest benchmark. A simple Random selection baseline performs significantly worse, establishing that learned saliency is non-trivial. Notably, a more sophisticated heuristic based on feature similarity to the text prompt performs catastrophically poorly, highlighting that token importance is a complex, non-linear function of the joint context that must be learned. When comparing optimization strategies, using Gumbel-Softmax instead of STE yields competitive but slightly inferior results. Lastly, reverting to a non-adaptive selection strategy where tokens are chosen from each modality's pool separately while maintaining the original token ratio (Per-modality) also leads to a noticeable performance drop. This confirms that the ability to dynamically arbitrate the token budget across modalities is critical for optimal performance.

**Analysis of Sync-RoPE Frequency Partitioning.** We validate the design of Sync-RoPE and its role in mitigating positional aliasing. As shown in Table 5, running our compressed model with the original TMRoPE ([16,24,24,0] split) results in the lowest performance among all variants, significantly underperforming the uncompressed full model. This drop is particularly pronounced in temporally sensitive sub-tasks like AV Event Alignment, confirming that standard RoPE fails in sparse sequences. Introducing Sync-RoPE by reallocating even a small number of low-frequency channels to the temporal dimension ([20,20,20,4] split) immediately recovers and even surpasses the full model's performance. This demonstrates the effectiveness and robustness of our spectral-based solution. While performance is stable across different partitions, our chosen [18,18,18,10] split yields the best overall results, striking

*Table 6.* **Ablation study on the number of CS2 encoder layers** ($N$). "Rel. Perf." is the average performance relative to the Full Model (100%). Our chosen setting ($N = 4$) is highlighted.

| Model | CS2 Layers ($N$) | Audio-Visual | | | | Video-Only | | | Rel. Perf. |
|---|---|---|---|---|---|---|---|---|---|
| | | World-Sense | Daily-Omni | Video-MME(A) | Avg | Video-MME | MLVU | Avg | |
| Full Model | – | 45.4 | 59.65 | 63.1 | 56.1 | 60.9 | 67.2 | 64.1 | 100.0% |
| Ours | 2 | 41.5 | 50.71 | 58.3 | 50.2 | 57.4 | 67.9 | 62.7 | 92.8% |
| Ours | 3 | 41.4 | 51.04 | 58.6 | 50.3 | 57.5 | 68.0 | 62.8 | 93.0% |
| Ours | 4 | 45.0 | 60.65 | 60.7 | 55.5 | 58.6 | 68.3 | 63.5 | **99.0%** |

an optimal balance between high-frequency local precision and low-frequency long-range stability.

**Analysis of CS2 Encoder Depth.** We investigate the impact of the Cross-Modal Encoder (CS2) depth ($N$). The depth $N$ is gradually decreased during the experiments, starting from the setting $N = 4$ in Section 4.1.

The results in Table 6 reveal an interesting capacity threshold: performance deteriorates significantly at shallower depths ($N = 2, 3$). We hypothesize that these configurations sit in a "capacity trap": the encoder attempts complex, global cross-modal modeling but lacks the sufficient depth to do so correctly, resulting in confused representations. In contrast, performance significantly improves at $N = 4$, where the model achieves 99.0% of the full model's performance. This confirms that $N = 4$ provides the necessary minimum capacity to unlock robust cross-modal alignment, validating our design choice and aligning with precedents in similar architectures (Zhang et al., 2025b).

### 4.5. Visualization

We provide visualizations to demonstrate the core mechanisms and effectiveness of *EchoingPixels*. We show that our method (1) retains fine-grained perceptual details, and (2) adaptively allocates the token budget. Only visual tokens are visualized, as audio features inherently lack intuitive visual correspondences.

**Retaining Fine-Grained Perception** Figure 6 confirms that critical details survive aggressive compression. Specifically, the model maintains temporal precision by accurately linking video segments to speech (Figure 6a) and preserves fine-grained visual cues—such as OCR tokens required for detailed QA (Figure 6b).

**Adaptive Budget Allocation** Figure 7 validates the efficacy of our adaptive combined pooling mechanism. Instead of fixed ratios, the model dynamically shifts budgets based on cross-modal density: prioritizing audio tokens during static visual scenes while reallocating resources to video during dynamic sequences.

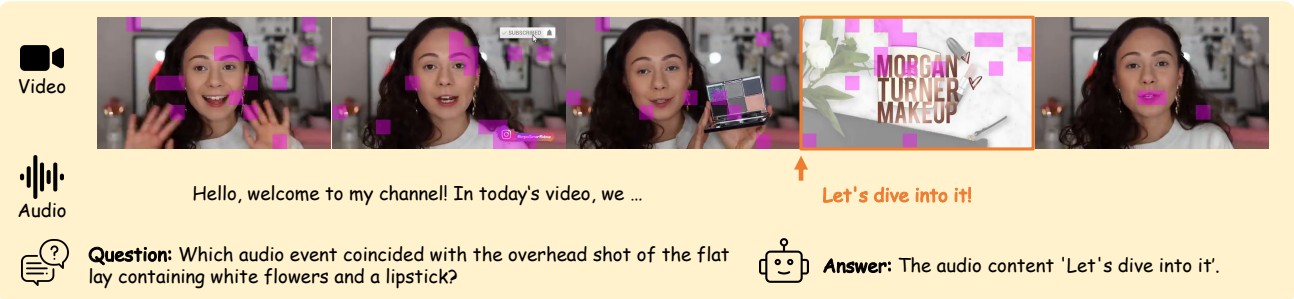

*(a)* Temporal Relationship: The model correctly aligns the audio "Let's dive into it!" with the corresponding product shot.

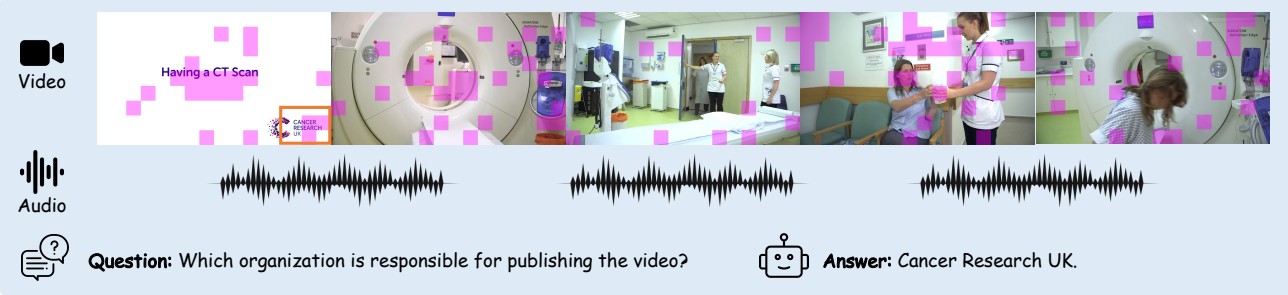

*(b)* OCR/Detail: The model correctly preserves sparse tokens on the small text "Cancer Research UK".

*Figure 6.* **Fine-grained perception after compression.** Selected tokens are highlighted.

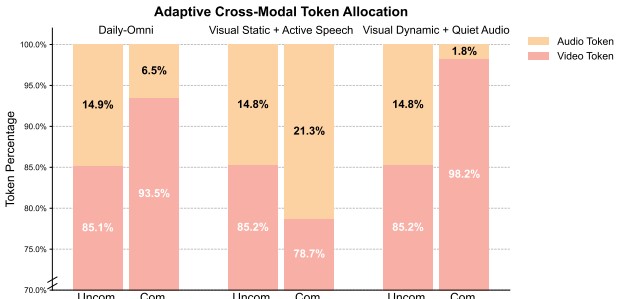

*Figure 7.* **Adaptive token allocation from the combined pool.** The model dynamically shifts the token budget from video to audio when video is static (middle) and from audio to video when audio is quiet (right). "Uncom." and "Com." indicate uncompressed and compressed, respectively.

## 5. Conclusion

We tackle the prohibitive computational cost of AV-LLMs by addressing a fundamental challenge in sparse sequence modeling. We systematically analyze Positional Aliasing, a previously overlooked phenomenon where aggressive token reduction corrupts standard positional encodings and degrades temporal understanding. To resolve this, we introduce *EchoingPixels*, a framework for aliasing-resistant joint token reduction. The framework features two key innovations: Sync-RoPE, a mathematically-grounded positional encoding that acts as a spectral low-pass filter to prevent aliasing, and the Cross-Modal Semantic Sieve (CS2), an efficient extractive module that selects salient tokens from a unified audio-visual stream. Extensive experiments demonstrate that *EchoingPixels* achieves performance comparable to the full model using only 5-20% of the original tokens.

## Limitations

Sync-RoPE is derived for RoPE, as it relies on the frequency-domain interpretation of rotary phases. While this covers most modern LLM backbones (Touvron et al., 2023; Bai et al., 2023), architectures using learnable absolute positional embeddings or relative position biases do not admit a direct frequency-band reallocation. Although the aliasing problem generalizes to any positional scheme pre-trained on dense sequences, extending aliasing-resistant corrections beyond RoPE remains an open direction for future work.

## Acknowledgements

This project was supported by NSFC Project (No. 62522206) and Ant Group Research Intern Program.

## Impact Statement

This paper presents work whose goal is to advance the field of Machine Learning. There are many potential societal consequences of our work, none which we feel must be specifically highlighted here.

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

## A. Quantitative Analysis of Positional Aliasing

To demonstrate the severity of positional aliasing, consider a typical setting where the model operates under a **90% token reduction rate** (i.e., retaining 10% of tokens). This implies an effective sampling interval of $T_s \approx 10$. According to the Nyquist sampling theorem (Shannon, 2006), the maximum recoverable frequency is $\theta_{\text{limit}} = \pi/T_s \approx 0.314$.

We examine the original TMRoPE configuration in Qwen2.5-Omni (Xu et al., 2025a), where the first 16 frequency bands ($j \in [1, 16]$) are allocated to temporal modeling, $j \in [17, 40]$ to spatial height, and $j \in [41, 64]$ to spatial width. With $\theta_{\text{base}} = 10^6$ and $d = 128$, the frequency of the $j$-th channel is $\theta_j = 1000000^{-2(j-1)/128}$.

We first demonstrate temporal channels ($j \in [1, 16]$) violate both conditions:

- **Immediate Aliasing ($j = 1 \sim 6$):** Frequencies range from $\theta_1 = 1.0$ to $\theta_6 \approx 0.34 > 0.314$. These channels violate the Nyquist limit immediately, degenerating into aliased noise that conflates adjacent and distant timestamps.

- **Periodicity Ambiguity ($j = 7 \sim 16$):** While $\theta_7 \approx 0.27 < 0.314$ nominally satisfies the Nyquist criterion, these channels suffer from limited wavelength coverage. At $j = 16$, the wavelength is only $\lambda_{16} = 2\pi/\theta_{16} \approx 160$ tokens; in a 2000-token sequence, the position embedding wraps around more than 12 times, introducing severe periodicity ambiguity that hinders global temporal reasoning.

We then demonstrate spatial channels ($j \in [17, 64]$) already satisfy both conditions:

- **No Aliasing:** $\theta_{17} \approx 0.0316 \ll 0.314$, and all subsequent $j > 17$ satisfy $\theta_j \leq \pi/T_s$.

- **No Periodicity Ambiguity:** The maximum positional ID difference along a spatial axis equals the number of tokens per row or column. With Qwen2.5-Omni's patch size of 28, even at 4096px resolution this is at most $4096/28 \approx 147$ tokens. Since $\lambda_{17} \approx 199 > 147$ and $\lambda_j > \lambda_{17}$ for all $j > 17$, no wrapping occurs.

Therefore, spatial channels require no correction, and Sync-RoPE selectively reallocates only the temporal channels. In contrast, Sync-RoPE reassigns temporal modeling to ultra-low-frequency channels ($j \in [55, 64]$) with magnitudes $\ll \pi/10$, guaranteeing both anti-aliasing and global monotonicity.

## B. More Experimental Settings

### B.1. Implementation Details

Training uses ms-swift (Zhao et al., 2024). We train IntraModal on the same data as our method. For video, the patch size is 14x14 pixels, but adjacent 2x2 tokens are merged before entering the LLM, so the patch size for the tokens seen by the LLM is 28x28. In both training and evaluation, we set the maximum resolution to 105369 following ms-swift's default resolution, resulting in at most 144 visual tokens per frame. For audio, it converts each second of audio into 25 tokens. Qwen2.5-Omni's default FPS is 2, but convolution is subsequently used to merge adjacent frames, so the LLM module receives an FPS of 1 by default. In summary, a video is converted to a maximum of 169 tokens per second, and a half-minute video is converted to over 5000 tokens.

### B.2. Baseline Details

Here are the details of the baseline implementations for comparison. In FastV, the filtering layer $K = 2$. In PyramidDrop, the LLM is divided into $S = 4$ stages, and tokens are removed at the end of each stage at a budget of $\lambda = 0.2$. In OmniZip, the ratios of audio and video tokens retained are both 0.2, and other parameters are consistent with its open-source settings.

### B.3. Benchmark Details

The Daily-Omni Benchmark (Zhou et al., 2025b) is used in experiments to analyze the model's performance because it provides a wide range of audio-visual task categories with detailed classifications, and can systematically evaluate the model's performance across different dimensions. A detailed introduction to each category is provided here, based on Daily-Omni's original paper. It contains multi-choice questions covering the following types: (1) AV Event Alignment: Questions to determine which audio and visual events occurred simultaneously with each other; (2) Event Sequence:

*Table 7.* **Comparison with FastV-tuned.** All methods use Qwen2.5-Omni-3B at 20% token budget. Rel. Perf. is relative to the full model.

| Method | WorldSense | Daily-Omni | Video-MME w/ audio | Video-MME w/o audio | MLVU | Rel. Perf. |
|---|---|---|---|---|---|---|
| FastV | 40.1 | 49.87 | 51.9 | 51.8 | 56.3 | 84.6% |
| FastV-tuned | 43.4 | 55.3 | 58.1 | 53.6 | 61.3 | 91.9% |
| *EchoingPixels* | 45.0 | 60.65 | 60.7 | 58.6 | 68.3 | **99.0%** |

*Table 8.* **Comparison of positional encoding variants on Daily-Omni.** All methods use Qwen2.5-Omni-3B at 20% token budget.

| Method | AV Align | Comparative | Context | Sequence | Inference | Reason | Overall |
|---|---|---|---|---|---|---|---|
| TMRoPE | 48.58 | 60.30 | 53.89 | 54.58 | 69.48 | 68.00 | 57.98 |
| TMRoPE-I | 50.42 | 63.36 | 55.44 | 50.98 | 68.83 | 69.14 | 57.89 |
| Sync-RoPE (Ours) | **52.10** | **61.83** | **59.07** | **57.19** | **72.73** | 68.57 | **60.65** |

Questions to determine the temporal sequence of visual and audio events in the video; (3) Reasoning: Questions to explain the cause or reason behind the occurrence of a visual or audio event in the video; (4) Inference: Questions to speculate on information not explicitly presented in the video; (5) Comparative: Questions to compare the similarity or difference between the audio and visual information of two or more events in the video; (6) Context Understanding: Questions to determine the contextual information surrounding a specific event in the video.

### B.4. Setup and Derivation for Figure 4

Figure 4 visualizes positional aliasing as a function of true token distance. For an adjacent token pair with interval $\Delta_s$, RoPE encodes a phase shift $\phi_j = \Delta_s \theta_j$ per frequency channel $j$, where $\theta_j = \theta_{\text{base}}^{-2(j-1)/d}$. Under aggressive downsampling, $\Delta_s$ becomes large, and $\phi_j$ may exceed $2\pi$, causing phase wrapping: the recovered phase collapses to $\tilde{\phi}_j = (\Delta_s \theta_j) \bmod 2\pi$. We then compute a per-channel wrapped-distance proxy:

$$\hat{\Delta}_j = \tilde{\phi}_j \, / \, \theta_j, \tag{8}$$

and plot its mean and standard deviation across temporal channels as the predicted distance. Under TMRoPE, phase wrapping causes large true distances to appear spuriously small, whereas Sync-RoPE, by reassigning temporal modeling to ultra-low-frequency channels where $\phi_j \ll 2\pi$, maintains a near-ideal linear relationship between predicted and actual distance.

## C. More Experimental Results

### C.1. Comparison with a Fine-Tuned Baseline

To further address potential concerns about training fairness, in addition to the IntraModal baseline, we also evaluate a fine-tuned variant of FastV (FastV-tuned) with the same training data as *EchoingPixels*. Results are reported in Table 7.

Fine-tuning narrows the gap between FastV and *EchoingPixels* (84.6% → 91.9%), yet a substantial margin remains (7.1 points), indicating that the performance gains of *EchoingPixels* stem from its design rather than training alone.

### C.2. Comparison with an Interleaved Frequency Variant

To further validate this design choice, we compare against TMRoPE-I, an interleaved $[t, h, w]$ frequency variant inspired by Qwen3-VL (Bai et al., 2025a), in which the three positional axes cycle across channels in a repeating pattern. Results on Daily-Omni are reported in Table 8.

TMRoPE-I underperforms Sync-RoPE (57.89 vs. 60.65). We attribute this to distribution shift: TMRoPE-I departs substantially from the TMRoPE scheme used during Qwen2.5-Omni's large-scale pre-training, making fine-tuning adaptation with only 390k samples harder. This result supports our conservative design of modifying only the temporal channels which are theoretically broken, while leaving the already-correct spatial channels intact.

*Table 9.* **Detailed latency breakdown (in ms) of the forward pass.** This table provides the detailed component-level average latency for the LLM forward pass of Qwen2.5-Omni-3B on Daily-Omni, supporting the analysis in the main text's Sec 4.3 Efficiency Analysis.

| Model | Token Budget | Cross-Modal Enc. | Selection | Decoder | Total Fwd |
|---|---|---|---|---|---|
| Full Model | 100% | - | - | 516.6 | 517.1 |
| Ours | 20% | 134.3 | 2.1 | 94.2 | 231.7 |
| Ours | 10% | 134.2 | 2.0 | 56.7 | 194.0 |
| Ours | 5% | 134.1 | 2.1 | 37.5 | 174.8 |

*Table 10.* **Latency Breakdown (in ms) across video durations.** All experiments use Qwen2.5-Omni-3B at 20% token budget. "Orig. Total" denotes the LLM forward latency; "Ours Total" sums Cross-Modal Enc., Selection, and Decoder.

| Duration (s) | Orig. Total | Cross-Modal Enc. | Selection | Decoder | Ours Total | Speedup |
|---|---|---|---|---|---|---|
| 5 | 54.4 | 7.2 | 1.0 | 19.6 | 27.8 | 2.0× |
| 10 | 102.6 | 15.2 | 1.1 | 30.1 | 46.4 | 2.2× |
| 20 | 201.0 | 37.7 | 1.2 | 47.3 | 86.2 | 2.3× |
| 30 | 314.3 | 69.9 | 1.4 | 66.1 | 137.4 | 2.3× |
| 60 | 730.1 | 219.0 | 4.8 | 124.8 | 348.6 | 2.1× |

## C.3. Detailed Efficiency Analysis

We break down the detailed average latency of the LLM forward pass on Daily-Omni in Table 9, clearly demonstrating the efficiency trade-offs of the *EchoingPixels* framework.

The baseline model Qwen2.5-Omni-3B has an LLM forward time of 517.1 ms, almost entirely consumed by its Decoder (516.6 ms). Our method introduces the overhead of two new modules: the Cross-Modal Encoder and the Selection operation. As shown in the table, this overhead is only one-quarter of the baseline model's decoder latency and is very stable, remaining at approximately 136 ms across all compression ratios. Therefore, our approach makes an extremely cost-effective trade-off: taking a 10% budget as an example, on the one hand, we "invest" 1/4 of the Decoder's fixed latency of 136.2ms; on the other hand, in return, we improve the Decoder's speed by 9.1 times, compressing it from 516.6ms to 56.7ms. Ultimately, the overall speed is improved by 2.67 times, strongly demonstrating that our CS2 module is a high-yield investment.

Table 10 further reports the latency breakdown across video durations ranging from 5 to 60 seconds-a range representative of common production scenarios such as content moderation and ad analysis. Across all durations, *EchoingPixels* achieves a consistent 2.0-2.3× speedup. This supports that the CS2 overhead is well amortized over the decoder savings.

## C.4. More Visualization

Figure 8 illustrates how our CS2 captures cross-modal synergies. In Figure 8a, the model uses the early audio cue "On your left" to identify the importance of the later visual portal. Conversely, in Figure 8b, the bidirectional context allows the **later** audio "What did you do?" to identify the **preceding** visual snap as the salient event. This holistic, non-causal understanding is impossible for standard baselines.

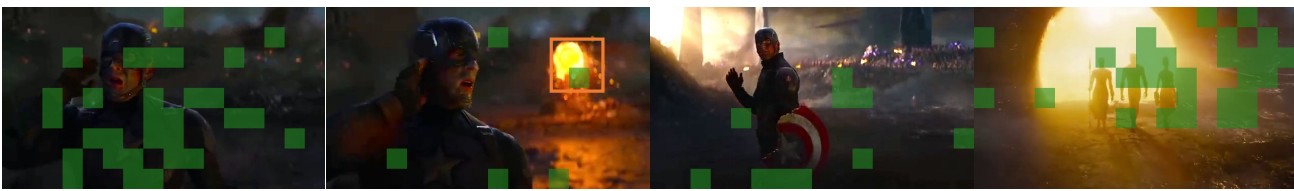

Captain, it's Sam. Can you hear me? On your left.

*(a)* An early audio cue ("On your left") guides the selection of later visual tokens (the portal).

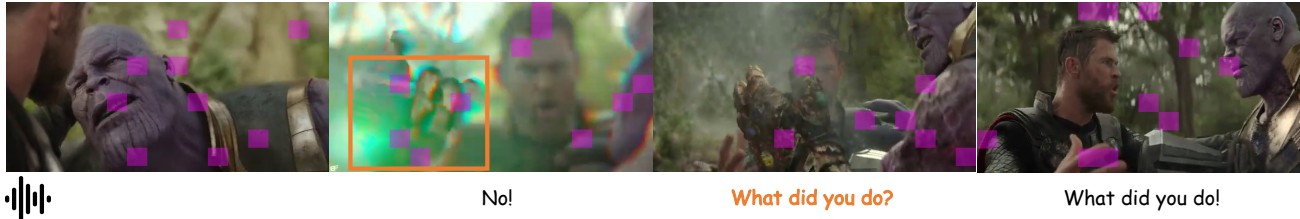

No!     What did you do?     What did you do!

*(b)* A **later** audio query ("What did you do?") directs the selection to a **prior** visual event (Thanos's snap).

*Figure 8.* **Demonstration of bidirectional and cross-modal selection.** Our CS2 holistically links audio-visual cues across time, which is impossible for causal or intra-modal methods. Selected tokens are highlighted (colors adjusted for clarity).

