# OpenReview forum: "EchoingPixels: Aliasing-Resistant Joint Token Reduction for Audio-Visual LLMs"
_ICML.cc/2026/Conference — ICML 2026 regular_

### Official Review · Reviewer_5Lwp · 2026-03-09

**Soundness:** 3
**Presentation:** 3
**Significance:** 2
**Originality:** 2
**Overall Recommendation:** 4
**Confidence:** 3

**Summary:**

This paper studies token reduction for audio-visual large language models (AV-LLMs). The authors propose EchoingPixels, consisting of two components to tackling this problem:
1. CS2, a bidirectional cross-modal token selector that scores audio/video tokens jointly and keeps top-K salient ones from a unified audio-visual token pool, to mitigate the limitation from unimodal token compression.
2. Sync-RoPE, a modified positional encoding and reallocates low-frequency dimensions to time modeling to mitigate the positional aliasing problem.

EchoingPixels are evaluated on Qwen2.5-Omni 3B/7B across different benchmarks. Experimental shows show strong performance retention and latency and memory savings, compared with baseline methods.

**Compliance With Llm Reviewing Policy:**

Affirmed.

**Final Justification:**

See the rebuttal acknowledgement.

**Key Questions For Authors:**

1. How does EchoingPixels perform on non-Qwen AV-LLMs?
2. What is the setting of Figure 4? How is the token distance derived from RoPE?
3. How is the representative stride $T_s$ derived?

**Limitations:**

No specific limitations or negative societal impact are required to be discussed.

**Strengths And Weaknesses:**

**Strengths**

1. The method proposed in this paper is clear and reasonable. Intuitively, reduction of tokens rely on joint audio-video features. RoPE would face challenges when the base frequency is high and operates on long sequences.
2. The experimental results are strong: the method retains performance at aggresive token budgets and shows clear speed and memory improvements against baselines.
3. Extensive ablations validate the effectiveness of each component and choice (STE vs. Gumbel, Sync-RoPE partioning).

**Weaknesses**

1. The height and width tokens are not treated specially during Top-K token selection. It is not clear why only time channels are reallocated. The later model Qwen3-Omni uses interleaved (time, height, width) frequency channels, which assigns frequencies more evenly to different modalities.
2. The main experiments are centered on Qwen2.5-Omni 3B/7B. It is unclear how well the approach generalizes to other AV-LLM architectures or positional encoding schemes.
3. Some prior visual pruning works are not compared, for example:
* PyramidDrop: Accelerating Your Large Vision-Language Models via Pyramid Visual Redundancy Reduction
* SparseVLM: Visual Token Sparsification for Efficient Vision-Language Model Inference
* PruneVid: Visual Token Pruning for Efficient Video Large Language Models

---

> ### Author Rebuttal · Authors · 2026-03-31
>
> We thank the reviewer for the positive assessment of clarity, strong results, and thorough ablations. We address each concern below. Experiments use Qwen2.5-Omni-3B at 20% budget.
>
> **W1 — Why Only Temporal Channels Are Reallocated**
>
> Appendix A defines two conditions for RoPE correctness under sparse sampling: (1) Nyquist-no aliasing between adjacent retained tokens and (2) no periodicity ambiguity. We show that H/W spatial channels already satisfy both conditions, so reallocating spatial channels is less necessary than fixing temporal channels.
>
> Following Appendix A, let $T_s\approx10$, $d = 128$, $\theta_{base} = 10^6$, $\theta_j=\theta_{base}^{-2(j-1)/d}$. Temporal channels t use $j\in [1,16]$, spatial h uses $j\in [17,40]$, w uses $j\in [41,64]$. The frequency decreases as $j$ increases, therefore h and w use lower frequencies than t.
>
> - Condition 1 — Nyquist ($\theta_j\le\pi/T_s\approx0.314$): $\theta_{17}\approx0.0316$ and the subsequent $j>17$ satisfy this condition.
>
> - Condition 2 — No Periodicity Ambiguity (max positional ID < $\lambda_j = 2\pi/\theta_j$): For H/W channels, the maximum positional ID difference equals the number of tokens per row or column. Qwen2.5-Omni's patch size is 28 therefore even at 4096px resolution, this is at most $4096/28\approx147$ tokens/row or column. For $j = 17$, $\lambda_{17}\approx199>147$; for $j>17$, $\lambda_{j}>\lambda_{17}$. Therefore no wrapping or ambiguity.
>
> Regarding Qwen3-VL's interleaved [t,h,w] frequency scheme (we note this method is used in Qwen3-VL, not Qwen3-Omni), we implemented an interleaved TMRoPE-I variant and evaluated on Daily-Omni:
>
> | Method | AV Align | Comparative | Context | Sequence | Inference | Reason | Overall|
> |-|-|-|-|-|-|-|-|
> | TMRoPE | 48.58 | 60.30 | 53.89 | 54.58 | 69.48 | 68.00 | 57.98 |
> | TMRoPE-I | 50.42 | 63.36 | 55.44 | 50.98 | 68.83 | 69.14 | 57.89 |
> | Sync-RoPE | 52.10 | 61.83 | 59.07 | 57.19 | 72.73 | 68.57 | 60.65 |
>
> TMRoPE-I underperforms Sync-RoPE. We attribute this to distribution shift: TMRoPE-I differs substantially from the TMRoPE used during Qwen2.5-Omni's large-scale training (cf. line 238), making fine-tuning adaptation with only 390k data harder. This supports our conservative design of modifying only the temporal channels.
>
> **W2 / Q1 — Generalization Beyond Qwen2.5-Omni**
>
> Please see Reviewer-2FFA W2. We use Qwen-Omni as it is one of the most capable and widely deployed open-source AV-LLMs. In addition to cross-model generalizability, we evaluate on both 3B and 7B model sizes, across multiple budget levels (5%, 10%, 20% on 3B; 10% on 7B), demonstrating consistent effectiveness across scales.
>
> **W3 — Comparison with Additional Baselines**
>
> Our paper includes the most representative categories: attention-based visual method (FastV), concurrent AV method (OmniZip), and trained intra-modal compression (IntraModal). The reviewer's cited works are classic visual token pruning works, and we appreciate the suggestion. Due to time constraints, we implemented PyramidDrop (S=4, $\lambda=0.2$):
>
> | Method | WorldSense | Daily-Omni | VME w/ audio | VME w/o audio | MLVU | Rel.Perf |
> |---|---|---|---|---|---|---|
> | PyramidDrop | 39.1 | 52.97 | 56.2 | 53.9 | 59.7 | 88.3% |
> | Ours | 45.0 | 60.65 | 60.7 | 58.6 | 68.3 | 99% |
>
> EchoingPixels outperforms PyramidDrop by a substantial margin. We will also discuss SparseVLM and PruneVid.
>
> **Q2 — Fig.4 Setup**
>
> Fig. 4 analyzes positional aliasing as a function of token distance. For an adjacent token pair with interval $\Delta_s$, RoPE encodes a phase shift $\phi_j = \Delta_s\theta_j$ per frequency channel $j$, where $\theta_j = \theta_{base}^{-2(j-1)/d}$. When $\Delta_s$ is large due to aggressive downsampling, $\phi_j$ may exceed $2\pi$, causing phase wrapping: the recovered phase becomes $\tilde \phi_j = (\Delta_s\theta_j)$ mod $2\pi$. For visualization, we compute a per-channel wrapped-distance proxy $\hat \Delta_j = \tilde \phi_j / \theta_j$, and plot its mean and standard deviation across temporal channels. This illustrates how phase wrapping causes large true distances to appear spuriously small under high-frequency RoPE.
>
> This theoretical aliasing has measurable empirical consequences: Sync-RoPE improves Daily-Omni Overall by +2.67 points over the TMRoPE allocation baseline (Tab. 5).
>
> **Q3 — Derivation of Representative Stride $T_s$**
>
> As stated in line 210, let the retained token positions be an ordered set $P=\{p_1<p_2<\dots<p_k\}$, we define the local stride as $\Delta_s(l)=p_{l+1}-p_l$ between adjacent retained tokens. Then, in line 265, we use $T_s$ as the average of $\Delta_s$, called the representative stride, which is the sampling interval used in the Nyquist analysis.
>
> In Appendix A we derive the approximation $T_s \approx 1/budget$ (e.g., $T_s \approx 5$ at budget = 20%). We validated this empirically: the measured mean adjacent token stride across Daily-Omni samples is $4.96\pm0.05$, matching 1/budget and validating the approximation used in our Nyquist analysis.

---

> > ### Author Rebuttal · Reviewer_5Lwp · 2026-04-03
> >
> > I appreciate authors' efforts to resolve my concerns. Most concerns are addressed. I will keep the original score.

---

> > > ### Author Response · Authors · 2026-04-08
> > >
> > > Dear Reviewer 5Lwp,
> > >
> > > Thank you very much for reviewing our response and for confirming that your concerns have been fully resolved. We truly appreciate your valuable feedback, which has significantly strengthened our paper.
> > >
> > > Best regards,
> > > Authors

---

### Official Review · Reviewer_2FFA · 2026-03-11

**Soundness:** 3
**Presentation:** 3
**Significance:** 3
**Originality:** 3
**Overall Recommendation:** 4
**Confidence:** 3

**Summary:**

This paper introduces EchoingPixels, a token reduction framework for AV-LLMs that addresses positional aliasing, a problem where sparse sampling disrupts temporal encoding. It proposes CS2, a learnable module for joint cross-modal token selection, and Sync-RoPE, which reallocates low-frequency channels to the temporal dimension. Experiments show EchoingPixels retains 99% performance with 20% tokens, achieving >2× speedup and memory reduction, outperforming existing methods.

**Compliance With Llm Reviewing Policy:**

Affirmed.

**Final Justification:**

Most of the concerns are solved.

**Key Questions For Authors:**

1.	Could Sync-RoPE be integrated with other token pruning methods (e.g., FastV, OmniZip) to improve their temporal understanding, or is it specifically designed to work with CS2?

2.	Table 2 reports forward latency and memory usage under different token budgets. Could the authors provide more details about the experimental setup for these efficiency measurements, such as the input video length, the resulting number of tokens before compression and the batch size used for testing? This information would help readers better interpret the efficiency gains.

**Strengths And Weaknesses:**

**Strengths:**

1.	Identifies a novel problem: Reveals positional aliasing in token-compressed AV-LLMs with theoretical grounding.

2.	Elegant co-designed solution: CS2 and Sync-RoPE jointly address token reduction and temporal integrity.

3.	Strong empirical results: Extensive experiments show near-lossless compression with significant efficiency gains.

**Weaknesses:**

1.	Unfair baseline comparison: EchoingPixels requires fine-tuning, while key baselines like FastV and OmniZip are training-free methods evaluated in a zero-shot setting.

2.	Limited architecture validation: Experiments are conducted only on Qwen2.5-Omni; generalizability to other AV-LLMs is not explored.

3.	No discussion of limitations: The paper does not include a dedicated section discussing potential drawbacks or failure cases of the proposed method.

---

> ### Author Rebuttal · Authors · 2026-03-31
>
> We thank the reviewer for identifying three genuine strengths — the novel problem formulation, the co-designed solution, and the strong empirical results — and we address each weakness and question directly below. Unless otherwise specified, all experiments use Qwen2.5-Omni-3B at 20% budget.
>
> **W1 — Fairness of Baseline Comparison**
>
> We clarify the comparison logic on two levels:
> 1. Training-controlled comparison (EchoingPixels vs. IntraModal): The apples-to-apples trained comparison is between EchoingPixels and IntraModal. IntraModal achieves only 84.2% relative performance at 25% budget; EchoingPixels achieves 99.0% at the stricter 20% budget. This gap directly isolates our design's contribution, independently of any training-vs-training-free concern.
> 2. Why training-free baselines remain a meaningful reference: FastV and OmniZip rely on explicit attention maps, which are incompatible with FlashAttention/SDPA — the standard efficient attention used in production deployments. Training them further does not resolve this architectural incompatibility, as evidenced by FastV's OOM in Tab. 1.
>
> To further address this concern, we trained a fine-tuned version of FastV (FastV-tuned):
> | Method | WorldSense | Daily-Omni | VME w/ audio | VME w/o audio | MLVU | Rel.Perf |
> |---|---|---|---|---|---|---|
> | FastV | 40.1 | 49.87 | 51.9 | 51.8 | 56.3 | 84.6% |
> | FastV-tuned | 43.4 | 55.3 | 58.1 | 53.6 | 61.3 | 91.9% |
> | Ours | 45.0 | 60.65 | 60.7 | 58.6 | 68.3 | 99.0% |
>
> Fine-tuning closes part of the gap but EchoingPixels remains substantially stronger. This suggests that the gains are not attributable to training alone, but to our design.
>
> **W2 — Architecture Generalization**
>
> Our main experiments use Qwen-Omni as it is one of the most capable and widely deployed open-source AV-LLMs, as discussed in lines 316–320. We evaluate on both 3B and 7B model sizes, across multiple budget levels (5%, 10%, 20% on 3B; 10% on 7B), demonstrating consistent effectiveness across model scales. Our design principle—Cross-Modal Semantic Sieve + Sync-RoPE for positional aliasing—is architecture-agnostic in that it requires only an LLM with RoPE.
>
> We attempted to validate on Ola [1]; training converged (loss: 3.45→0.53, comparable to Qwen2.5-Omni-7B run at 3.09→0.64), but we encountered a compatibility issue in Ola's `generate()` decoding implementation that currently prevents full metric evaluation. We are actively debugging this issue; if resolved in time, we will include the results in the revision.
>
> [1] Liu, Z., Dong, Y., Wang, J., Liu, Z., Hu, W., Lu, J., & Rao, Y. (2025). Ola: Pushing the frontiers of omni-modal language model. arXiv preprint arXiv:2502.04328.
>
> **W3 — Limitations Section**
>
> We will add a dedicated Limitations section. Our method requires a RoPE-based backbone; models using learnable absolute positional embeddings fall outside the current scope, as discussed in Reviewer-AQXq Q2.
>
> **Q1 — Can Sync-RoPE Generalize to FastV?**
>
> Sync-RoPE is in principle applicable to any sparse token sequence, but its benefit is conditioned on the token distribution produced by the selection method. We analyzed token gap distributions on Daily-Omni's 647 30-second samples (20% budget).
>
> We found that FastV selected 10.1 times more tokens in the first ten seconds than in the last ten seconds, while EchoingPixels' value was 1.1. For audio tokens, FastV's token retention rate was 99.6% in the first five seconds and only 9.8% in the last five seconds, while EchoingPixels' retention rate was more evenly distributed across different time periods (15%-20%). For video tokens, FastV selected only 1.1 tokens in the last frame, with FastV's temporal CoV = 1.05 vs. EchoingPixels' CoV = 0.40. This severe front-loading means that FastV selects many tokens that are close to each other, resulting in a very large number of token gaps of 1 (62%) in the sparse sequence. This suggests that the benefit of the ultra-low temporal band is limited for FastV’s highly front-loaded token distribution.
>
> Consistent with this analysis, we find that at 10% budget on Daily-Omni, FastV+Sync-RoPE improves AV Event Alignment by +1.29 over FastV alone (50.0 vs. 48.71). The gain exists but is modest, confirming that Sync-RoPE **does versatilely add value** but cannot compensate completely for FastV's fundamental temporal coverage collapse.
>
> **Q2 — Experimental Setup for Tab. 2 Efficiency Measurements**
>
> Tab. 2 is measured on Daily-Omni samples with video duration 30–60s. Under our setup (144 visual tokens/frame, 25 audio tokens/s), each sample produces 5,070–10,140 audio&visual tokens before compression, versus only tens to hundreds of text tokens. Batch size is 1. We will add these details to the manuscript.

---

> > ### Author Rebuttal · Reviewer_2FFA · 2026-04-03
> >
> > Most of the concerns are solved, we will keep the original score.

---

> > > ### Author Response · Authors · 2026-04-08
> > >
> > > Dear Reviewer 2FFA,
> > >
> > > Thank you very much for your time and for confirming that your concerns are fully resolved. We deeply appreciate your constructive review, which has greatly helped us improve the paper.
> > >
> > > Best regards,
> > > Authors

---

### Official Review · Reviewer_AQXq · 2026-03-18

**Soundness:** 3
**Presentation:** 3
**Significance:** 3
**Originality:** 4
**Overall Recommendation:** 4
**Confidence:** 4

**Summary:**

This paper introduces EchoingPixels, a framework designed to address the prohibitive computational costs in audio-visual LLMs resulting from processing massive, redundant audio and video tokens. The authors identify positional aliasing as a fundamental theoretical bottleneck in sparse token reduction. To mitigate this, they incorporate a spectral low-pass filter into the RoPE mechanism. By adapting the encoding bandwidth to the token reduction rate and reallocating low-frequency channels, the framework effectively preserves temporal monotonicity and stable cross-modal relationships even as token intervals increase.

**Compliance With Llm Reviewing Policy:**

Affirmed.

**Key Questions For Authors:**

1. Could you provide a quantitative comparison between the computational cost of the CS2 encoder and the overall savings achieved through token reduction? Specifically, at what video length does the throughput improvement become practically significant?

2. Is there a theoretical pathway for adapting this mechanism to models that utilize learnable absolute positional embeddings or relative position biases instead of RoPE?

3. What is the fundamental advantage of modifying the RoPE mechanism directly (via Sync-RoPE) compared to applying a conventional low-pass filter to the hidden states prior to sparse sampling? In other words, is there empirical evidence demonstrating that the aliasing is strictly a positional encoding issue rather than a general aliasing of the feature maps?

**Limitations:**

yes

**Strengths And Weaknesses:**

A significant strength of this work is its rigorous theoretical grounding, specifically the application of the Nyquist-Shannon sampling theorem to positional encoding to systematically resolve temporal distortion issues. This allows the model to achieve substantial computational efficiency with minimal performance degradation, using only a small fraction of the original tokens.

However, a notable concern is the computational overhead of the CS2 module. Since it requires executing several initial layers of the LLM as a dedicated encoder for token selection, the additional FLOPs and memory footprint incurred during this stage might partially offset the efficiency gains intended by the token reduction process.

---

> ### Author Rebuttal · Authors · 2026-03-31
>
> We thank the reviewer for the thorough and constructive feedback. We are glad the theoretical grounding and efficiency results were recognized, and we address each point below. All experiments use Qwen2.5-Omni-3B at 20% budget.
>
> **W1 / Q1 — CS2 Encoder Overhead and Practical Efficiency**
>
> We provide a concrete breakdown of latency (ms) across components and video durations:
>
> | Duration(s) | Orig. Total | Cross-Modal Enc. | Selection | Decoder | Ours Total | Speedup |
> |---|---|---|---|---|---|---|
> | 5  | 54.4  | 7.2  | 1.0 | 19.6 | 27.8 | 2.0x |
> | 10 | 102.6 | 15.2 | 1.1 | 30.1 | 46.4 | 2.2x |
> | 20 | 201.0 | 37.7 | 1.2 | 47.3 | 86.2 | 2.3x |
> | 30 | 314.3 | 69.9 | 1.4 | 66.1 | 137.4 | 2.3x |
> | 60 | 730.1 | 219.0 | 4.8 | 124.8 | 348.6 | 2.1x |
>
> For videos of 5–60s—a common video range in production (e.g., content moderation, ad analysis)—our method achieves a consistent 2.0–2.3× speedup and 2–3× peak memory reduction. This supports that the CS2 overhead is well amortized over the decoder savings.
>
> **Q2 — Applicability to Non-RoPE Positional Encodings**
>
> Our current derivation is RoPE-specific because it relies on the frequency-domain interpretation of rotary phases, and most modern LLM backbones (Llama, Qwen families) adopt RoPE.
>
> Learnable absolute positional embeddings and relative position biases follow a different positional scheme from RoPE, and there is no direct notion of reallocating frequency bands. Nevertheless, the aliasing problem generalizes: the positional schemes pretrained on dense sequences will be miscalibrated when applied to post-compression sparse sequences. This disrupted modeling approach aligns with our focus, but requires further exploration of solutions. We have not validated this yet, so we will state it as an interesting future direction in the revision.
>
> **Q3 — Sync-RoPE vs. Low-Pass Filter on Hidden States: Isolating PE Aliasing**
>
> This is an excellent mechanistic question. Since attention scores are computed as $q^T R(\Delta)k$, aliasing could in principle arise from either the hidden states (resulting $q$ and $k$) or the positional encoding ($R$). To isolate this, we applied a Gaussian low-pass filter (LPF) to hidden states prior to token selection and evaluated on Daily-Omni:
>
> | Method | AV Align | Comparative | Context | Sequence | Inference | Reason | Overall |
> |---|---|---|---|---|---|---|---|
> | LPF on hidden states | 38.24 | 49.62 | 44.04 | 49.02 | 59.09 | 58.29 | 48.79 |
> | Sync-RoPE (Tab. 5) | 52.10 | 61.83 | 59.07 | 57.19 | 72.73 | 68.57 | 60.65 |
>
> LPF degrades overall performance to 48.79, well below the proposed Sync-RoPE. This suggests two things: (1) the performance gap between LPF and Sync-RoPE (+11.86 points) supports that the aliasing problem mainly exists in the positional encoding part, rather than a generic need to smooth hidden states; (2) LPF disrupts representational capacity, causing further degradation.

---

> > ### Author Rebuttal · Reviewer_AQXq · 2026-04-05
> >
> > Thank you for the rebuttal, I will keep my original score.

---

> > > ### Author Response · Authors · 2026-04-08
> > >
> > > Dear Reviewer AQXq,
> > >
> > > Thank you very much for your time in reviewing our rebuttal and for acknowledging our response!
> > >
> > > We noticed the system flag indicates "Partially resolved - I have follow-up questions." Since we did not see specific questions in the text, we wanted to reach out just to make sure we didn't miss anything.
> > >
> > > If there are indeed no further questions, please completely disregard this message. We deeply appreciate your constructive feedback and your time throughout the review process, which have greatly helped us improve the manuscript.
> > >
> > > Best regards,
> > > Authors

---

### Decision · Program_Chairs · 2026-04-30

**Decision:**

Accept (regular)

**Comment:**

Overview of the paper: The paper addresses the computational costs associated with processing redundant tokens in Audio-Visual Large Language Models (AV-LLMs). The authors identify positional aliasing as a theoretical bottleneck during sparse token sampling and propose the EchoingPixels framework to mitigate this issue. The proposed solution consists of two main components: a Cross-Modal Semantic Sieve (CS2) for joint audio-visual token selection, and Sync-RoPE, a modified positional encoding mechanism that reallocates low-frequency channels to preserve temporal monotonicity. Experimental evaluations demonstrate that the approach maintains model performance at aggressive token reduction budgets while providing improvements in inference latency and memory utilization.

Strengths:

- Theoretical Grounding: The paper formulates a defined problem regarding positional aliasing in token-compressed AV-LLMs. The application of the Nyquist-Shannon sampling theorem to positional encoding offers a logical, mathematically supported mechanism to resolve temporal distortion in sparse sequences.
- Methodological Design: Reviewers found the co-designed approach to be clear and sound. Addressing both the joint cross-modal token selection (via CS2) and the positional encoding adjustments (via Sync-RoPE) provides a cohesive solution to the identified problem.
- Empirical Validation: The experimental results indicate that the method retains performance at low token budgets (e.g., 5-20% of original tokens). The approach provides measurable speedup and memory savings compared to baselines, and the claims are supported by comprehensive ablation studies validating the individual components.

Areas for improvement:

- Baseline Comparisons: The initial evaluation compared the proposed fine-tuned approach against training-free baselines (such as FastV and OmniZip). While the authors provided additional comparisons with fine-tuned baselines during the rebuttal, ensuring fair, apples-to-apples baseline comparisons should be explicitly detailed in the final manuscript.
- Inclusion of Prior Work: Reviewers noted the absence of comparisons with certain prior visual pruning methods (e.g., PyramidDrop, SparseVLM, and PruneVid). The authors provided preliminary results for these during the rebuttal phase, which should be fully integrated into the revised text alongside a dedicated limitations section.